



# True eddy accumulation - Part 2: Theory and experiment of the short-time eddy accumulation method

Anas Emad[1] and Lukas Siebicke[1]

[1]Bioclimatology, University of Göttingen, Büsgenweg 2, 37077 Göttingen, Germany

**Correspondence:** Anas Emad (anas.emad@uni-goettingen.de)

**Abstract.** A new variant of the eddy accumulation method for measuring atmospheric exchange is derived and a prototype sampler is evaluated. The new method, termed short-time eddy accumulation (STEA), overcomes the requirement of fixed accumulation intervals in the true eddy accumulation method (TEA) and enables the sampling system to run in a continuous flow-through mode. STEA enables adaptive time-varying accumulation intervals which improves the system's dynamic range and brings many advantages to flux measurement and calculation.

The STEA method was successfully implemented and deployed to measure $CO_2$ fluxes over an agricultural field in Braunschweig, Germany. The measured fluxes matched very well against a conventional EC system (slope of 1.04, $R^2$ of 0.86). We provide a detailed description of the setup and operation of the STEA system in the continuous flow-through mode, devise an empirical correction for the effect of buffer volumes, and describe the important considerations for the successful operation of the STEA method.

The STEA method reduces the bias and uncertainty in the measured fluxes and creates new ways to design eddy accumulation systems with finer control over sampling and accumulation. The results encourage the application of STEA for measuring fluxes of more challenging atmospheric constituents such as reactive species.

## 1 Introduction

Monitoring the exchange of trace gases and energy between the earth's surface and the atmosphere is a key problem in ecology and climate science. The eddy covariance method (EC) has become the standard method for estimating the flux density on the scale of plant canopies (Baldocchi, 2014; Hicks and Baldocchi, 2020). The flux in the EC method is calculated as the covariance between the vertical wind velocity and the scalar concentration. For this, EC requires the availability of high-frequency measurements of the vertical wind velocity and the concentration of the atmospheric constituent ($\geq$ 10 Hz). This requirement limits the EC method to few trace gases where fast-response gas analyzers are available. For constituents where only slow-response gas analyzers are available, several methods for measuring the fluxes exist (Rinne et al., 2021). Among these methods, the true eddy accumulation (TEA) (Desjardins, 1977) is the most direct and the closest to EC. TEA is formulated using similar principles and assumptions to the EC method. However, unlike EC, the TEA method requires the concentration measurements to be carried out once every averaging interval (30 minutes). For a long time, the development of the TEA method was hindered by the difficulty of fast air flow rate control and the strict operational requirements (Businger and Oncley,



1990; Hicks and McMillen, 1984). A recent improvement to the TEA method used a new type of mass flow controller, online coordinates rotation, and several online treatments of the signal to overcome important limitations of the method's applicability (Siebicke and Emad, 2019). The new system showed a good match with a reference eddy covariance system with coefficients of determination of up to 86% and a slope of 0.98. While this study demonstrated a successful proof-of-concept of TEA using modern sampling, it also showed that further research was required for continuous accumulation and long-term field operation, which we address with the current study.

The absence of high-frequency measurements of the scalar concentration creates unique challenges to the TEA method. The sampling decisions in TEA need to be done in real-time without complete knowledge of the wind statistics of the averaging interval. The problem of nonzero mean vertical wind velocity, a direct consequence of this limitation, is discussed in the accompanying paper (Emad and Siebicke, 2022).

Furthermore, the lack of high-frequency scalar measurements implies that sample accumulation needs to happen on a time scale similar to the flux averaging interval (30 to 60 minutes). Therefore, imposing a minimum limit on the sampling accumulation interval before the scalar concentration measurement can be conducted. This time limit imposes restrictive design considerations related to the size and function of sample accumulation reservoirs. It dictates that the sampling apparatus needs to accommodate a large dynamic range (up to $5\ \sigma_w$) to cover the range of wind velocities during the flux averaging interval (Hicks and McMillen, 1984). The minimum time limit is also problematic if the sampled scalar changes in concentration over time, e.g., reactive species. Additionally, the accumulation for long time intervals and the discontinuous nature of sample collection are particularly sensitive to instationary conditions in the accumulation apparatus (Siebicke and Emad, 2019). Additionally, the use of expandable bags in discrete sampling for the accumulation reservoirs has proven to be unreliable and prone to mechanical fatigue (Siebicke and Emad, 2019). Therefore, a more flexible approach is needed where the accumulation interval can be adapted to the requirements of the sampling system and the trace gas being measured.

In this paper, we address these limitations by developing a novel method for eddy accumulation and providing a prototype implementation of such a system. First, we derive a new eddy accumulation method, which we call the short-time eddy accumulation (STEA). STEA method enables the sample accumulation to be carried out on variable shorter intervals which brings many improvements in the TEA flux measurements including the ability to accumulate samples in a continuous flow-through mode. Next, we discuss the effect of using buffer volumes on the concentration measurements and develop an empirical correction for the use of buffer volumes. Finally, we show a prototype and experimental measurements for $CO_2$ fluxes using the newly developed STEA method in the flow-through mode and compare the measured fluxes to reference EC measurements. We discuss the advantages and steps required to carry out flux measurements using the STEA method, different constraints and operational requirements.

## 2 Theory

A detailed description of the TEA method derivation and assumptions is provided in the accompanying paper (Emad and Siebicke, 2022). Here we present a short overview to prepare for the short-time eddy accumulation derivation.





Under the assumptions of flow homogeneity and stationarity, the vertical exchange of the atmospheric scalar $c$ is the flux
across the measurement plane at height $h$, the flux $F_c$ is (Finnigan et al., 2003; Gu et al., 2012)

$$F_c = \overline{cw} \tag{1}$$

Here, $w$ is the vertical wind velocity $(\mathrm{m\,s^{-1}})$, $c$ is the scalar density $(\mathrm{mol\,m^{-3}})$, and the over-bar denote time averages that
follow Reynolds averaging rules.

The true eddy accumulation method is formulated by partitioning the average $\overline{wc}$ using the direction of the vertical wind
velocity. Therefore, we write the flux as the expected value of the random variable $wc$ conditional on the sign of the vertical
wind velocity, $\mathrm{sign}(w)$

$$\overline{wc} = \overline{w^{\uparrow}c^{\uparrow}}\,\mathrm{P}(w^{\uparrow}) + \overline{w^{\downarrow}c^{\downarrow}}\,\mathrm{P}(w^{\downarrow}) \tag{2}$$

Where the arrows denote the direction of the vertical wind velocity, $\uparrow$ for updraft, and $\downarrow$ for downdraft. $\mathrm{P}(w^{\uparrow\downarrow})$ is the
probability that the observed wind velocity is in the respective direction. The TEA method makes use of this simple partitioning
by physically realizing the terms $\overline{w^{\uparrow}c^{\uparrow}}$ and $\overline{w^{\downarrow}c^{\downarrow}}$ using sample accumulation instead of measuring individual realizations of $w$
and $c$. For the practical implementation of the TEA system, a parameter $A$ is necessary to relate the sampling flow rate to the
measured $w$.

## 2.1 Short-time eddy accumulation

The original formulation of the true eddy accumulation method requires the samples to be accumulated for the entire averaging
interval $\Delta t$ before the concentration measurement is ready for flux calculation. This limits the dynamic range and the flexibility
of the sampling system.

To achieve a higher dynamic range for the sampling system and realize a more robust flow-through eddy accumulation
system, we propose a modification to the TEA method where samples can be accumulated for a sequence of shorter intervals
$\tau_i$ that add up to the averaging period $\Delta t$.

This formulation can be achieved by applying the law of total expectation to the random variable $cw$ with respect to a
partitioning variable $Y$ that divides the averaging period $\Delta t$ into multiple non-overlapping partitions with the length $\tau_i$. This
partitioning scheme is applied individually to updarft and downdraft after partitioning with the direction of vertical wind
velocity. Therefore, we write the expectation of $\overline{c^{\uparrow}w^{\uparrow}}$

$$\overline{c^{\uparrow}w^{\uparrow}} = \overline{\left(\overline{(c^{\uparrow}w^{\uparrow})|Y}\right)} = \sum_i \overline{(c^{\uparrow}w^{\uparrow})|Y_i}\,\mathrm{P}(Y_i) \tag{3}$$

The previous equation is similarly valid for the downdraft flux $\overline{c^{\downarrow}w^{\downarrow}}$. The measured concentration during a short averaging
interval $i$, is given by





$$C_i = \frac{\overline{cw|Y_i}}{\overline{|w|}} \tag{4}$$

The probability of the short averaging interval can be obtained easily, $\mathrm{P}(Y_i) = \tau_i/\Delta t$.

$V_i$ is the volume accumualted during the short interval $i$, defined as

$$V_i = A_i \int_{t}^{t+\tau_i} |w|\, \mathrm{d}t \tag{5}$$

The concentration in either updraft or downdraft reservoirs for the averaging interval $\Delta t$ is the weighted mean of the short interval concentration measurements, $C_i$

$$C_{acc}^{\uparrow\downarrow} = \frac{1}{\overline{|w|}\Delta t} \sum_{i=1}^{i=j} C_i^{\uparrow\downarrow} \overline{|w_i|}\tau_i \tag{6}$$

We notice here that $\overline{|w_i|}\tau_i = V_i/A_i$ and $\overline{w}\Delta t = \sum_{i=1}^{i=j} V_i/A_i$.

The obtained $C_{acc}^{\uparrow}$ and $C_{acc}^{\downarrow}$ can be used to calculate the STEA flux (Emad and Siebicke, 2022)

$$F_{\mathrm{STEA}} = \frac{C_{acc}^{\uparrow} V^{\uparrow}\left(\overline{|w|} - \bar{w}\right) - C_{acc}^{\downarrow} V^{\downarrow}\left(\overline{|w|} + \bar{w}\right)}{\overline{|w|} - \alpha_c \bar{w}} \times \frac{\overline{|w|}}{V_{total}} \tag{7}$$

Where $F_{\mathrm{STEA}}$ is the kinematic flux density $(\mathrm{mol\,m\,s^{-1}})$. $C_{acc}^{\uparrow}$ and $C_{acc}^{\downarrow}$ are the mean molar densities $(\mathrm{mol\,m^{-3}})$ of the scalar $c$ in updraft and downdraft reservoirs for the whole accumulation period $\Delta t$ as calculated from Eq. 6. $V^{\uparrow}$ and $V^{\downarrow}$ are the accumulated sample volumes $(\mathrm{m^3})$ in updraft and downdraft reservoirs during the averaging period. It is important here to use $V^{\uparrow}$ and $V^{\downarrow}$ as $\overline{|w|}\Delta t^{\uparrow}$ since the parameter $A$ was not constant for different short intervals. $\overline{|w|}$ is the mean of the absolute vertical wind velocity $(\mathrm{m\,s^{-1}})$ during the averaging period. $\bar{w}$ is the mean of the vertical wind velocity. $\alpha_c$ is the transport asymmetry coefficient for the scalar $c$ (dimensionless).

## 2.2 Effect of buffer volumes

The short-time eddy accumulation method can be achieved in at least two ways, either using expandable buffer volumes (e.g., bags), which are emptied after each short interval measurement $C_i$ or using a flow-through system with rigid buffer volumes. The flow-through system has practical operational benefits but requires additional correction to reverse the effect of buffer volumes on the concentration signal. Buffer volumes act as low pass filters (Cescatti et al., 2016). They attenuate the magnitude of the high-frequency part and shift the phase of the signal. The buffer concentration at time step $n$ is dependent on the new input sample concentration and the buffer concentration from the previous step $y[n-1]$. Thus, the buffer volume concentration $y_n$ response to an input $C_i$ can be described with the difference equation





$$y_{[n]} = C_{i[n]} \dot{q}_i + (1 - \dot{q}_i) y_{[n-1]} \qquad (8)$$

where $\dot{q}$ is a dimensionless flow rate that is defined as the ratio between the sample mass to the total mass of air in the buffer volume, at each time step $n$

$$\dot{q}_n = \frac{V_i \, \rho_i}{V_b \, \rho_b} \qquad (9)$$

Where $V_i$ and $\rho_i$ are the volume and density of the accumulated sample during the interval, $i$, respectively. $V_b$ and $\rho_b$ are the volume and the air density of the air in the buffer volume, respectively. Equation (8) characterizes a first-order linear filter. The short accumulated samples in the STEA method are individually separated as they are forwarded to the gas analyzer, this discrete behavior is best modeled with a discrete-time system as shown in Eq. (8).

The system response is characterized by the dimensionless flow rate. The time constant of the system is defined as the
required time for the system to reach $1/e$ from a step increase and relates to $\dot{q}$ by (Taylor et al., 2013).

$$\tau = -\frac{\Delta s}{\ln(1 - \dot{q})} \qquad (10)$$

where $\Delta s$ is the length of the sampling interval.

Figure (1) shows the filter's magnitude and phase responses. The magnitude response $|H|$ plot shows how the magnitudes of different frequencies are attenuated, the smaller the dimensionless flow rate is, the larger the time constant is. Consequently,
the attenuation is stronger.

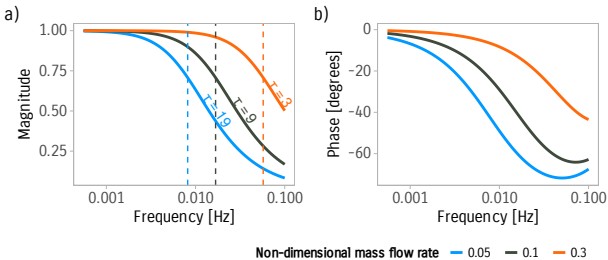

**Figure 1.** Frequency response for the first order linear filter used to model the buffer volumes for three different time constants. a) Magnitude response of the filter. Vertical dashed lines represent the cutoff frequencies for the respective time constants. b) Phase response of the filter.





## 2.3 Methods

### 2.3.1 Experimental site

Flux measurements were performed over a flat agricultural field of the Thünen Institute, located at 52.297 N, 10.449 E in Braunschweig, Germany. The site has an altitude of 76 m above sea level. During the measurement period, the fields south and north of the tower were planted with oats and corn, respectively. Both crops had a similar height of approximately 50 cm above the ground at the start of the comparison period.

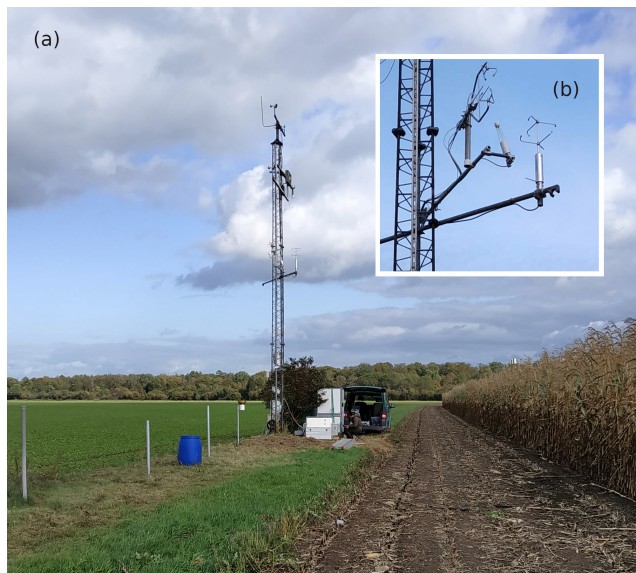

**Figure 2.** Photograph of the experimental field site showing the measurement tower **(a)** and a close up on the flux instruments mounted on the tower **(b)**.

### 2.3.2 Experiment period

Fluxes were measured throughout the year 2020. We selected six weeks of good quality in summer based on instrument performance and weather conditions, spanning from 18 June 2020 to 31 July 2020 to compare the different methods.

### 2.3.3 Instruments

EC and STEA measurement complexes were mounted at 5 m height above the ground (Fig. 2). The instruments used in the experiment for flux measurements and data analysis are listed in Table 1. Meteorological variables were logged using a Sutron 9210 XLite logger (Sterling, USA). All the raw data needed for flux processing were synchronized on the STEA computer and remote servers for real-time processing.





**Table 1.** Variables and instruments. Manufacturer key: METEK GmbH (Elmshorn, Germany), LI-COR Environmental Inc. (Lincoln, Nebraska, USA), LGR, (Los Gatos Research Inc., USA), Bosch (Bosch Sensortec GmbH, Germany), Vaisala (Helsinki, Finland), Kipp & Zonen (Delft - The Netherlands), Delta-T Devices Ltd (UK), Stevens Water Monitoring Systems, Inc (Oregon, USA), Texas Electronics (Dallas, USA)

| Variable | Sensor | Manuf. | Method | Freq. |
|---|---|---|---|---|
| Wind $u, v, w$ | uSonic-3 Omni H | METEK | EC | 20 Hz |
| Sonic temp. Ts | uSonic-3 Omni H | METEK | EC | 20 Hz |
| Wind $u, v, w$ | uSonic-3 Class A | METEK | TEA | 10 Hz |
| Sonic temp. Ts | uSonic-3 Class A | METEK | TEA | 10 Hz |
| $CO_2$ density | LI-7500A | LI-COR | EC | 10 Hz |
| $H_2O$ density | LI-7500A | LI-COR | EC | 10 Hz |
| $CO_2$ ppm | FGGA-24r-EP | LGR | TEA | 1 Hz |
| $H_2O$ ppm | FGGA-24r-EP | LGR | TEA | 1 Hz |
| $CH_4$ ppm | FGGA-24r-EP | LGR | TEA | 1 Hz |
| Air pressure P | BME280 | Bosch | TEA | 50 Hz |
| Air temperature | BME280 | Bosch | TEA | 50 Hz |
| Air humidity | HMP155 | Vaisala | Meteo | 10min |
| Air temperature | HMP155 | Vaisala | Meteo | 10min |
| Net radiation | CNR4 | KIPP | Meteo | 10min |
| Global radiation | BF5 | DELTA-T | Meteo | 10min |
| Soil heat flux | HFP01 | LI-COR | Meteo | 10min |
| Soil moisture | SDI-12 | Stevens | Meteo | 10min |
| Precipitation | TR-525M | Texas Elec. | Meteo | 10min |

The EC system comprised a dedicated sonic anemometer (uSonic-3 Omni H) and an open-path infra-red gas analyzer (IRGA). Wind and scalar density data were acquired at 20 Hz frequency. Relative to the Class-A sonic anemometer used for STEA, the northward, eastward, and vertical separation of the IRGA was $-17$ cm, 26 cm, and $-15$ cm, respectively. The Class-A sonic had a north offset azimuth of $90°$ degrees. Relative to the Omni-sonic anemometer used for EC, the northward, eastward, and vertical separation of the IRGA was 20 cm, $-15.3$ cm, and $-20$ cm. The north offset of the Omni-sonic was

$169°$ degrees.

## 2.4   STEA system description

The STEA system used in the experiment is based on an earlier system of Siebicke and Emad (2019). The new system used the same mass flow controllers and shared most of the operating software. It has, however, several differences and improvements. One major difference is the use of fixed stainless steel buffer volumes instead of expandable bags. The system was developed

initially as a hybrid TEA-EC method to run the TEA method in a continuous flow-through mode (Siebicke, 2016). The system was set up to operate in the STEA continuous flow-through mode. A constant duration for the short intervals ($\tau_i$) equal to one minute was used. The STEA system is comprised of two identical sampling lines, one for updrafts and one for downdrafts. Each of the sampling lines has two rigid buffer volumes in a sequence connected using 6 mm Teflon tube (Fig. 3).





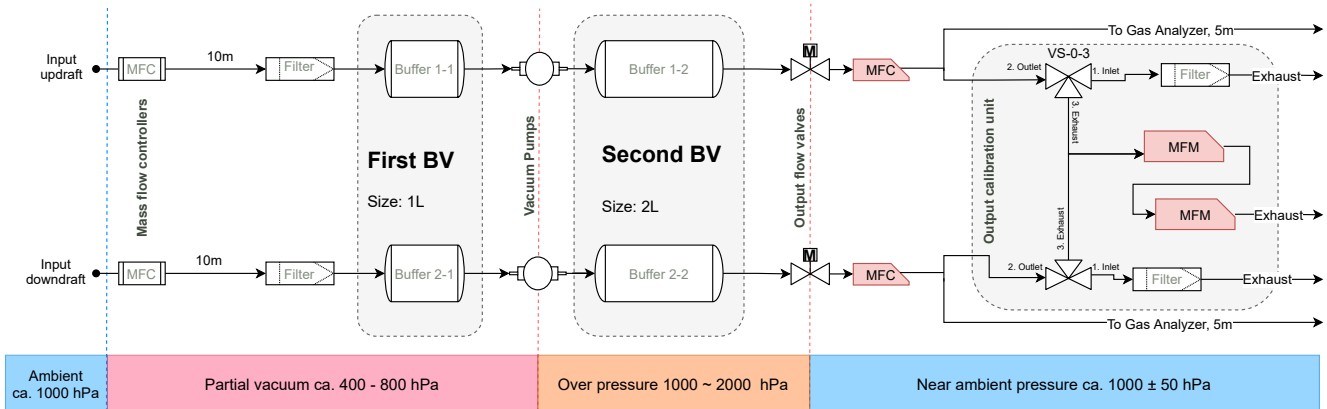

**Figure 3.** Functional and pneumatic schematic of the implemented flow-through STEA system showing components, layout, properties, and operation conditions. Air samples are collected at the input and travel in distinct sampling lines for updrafts and downdrafts. Samples travel through tubes (lengths are shown), through filters, are then collected into two sets of buffer volumes shown here as *First BV* and *Second BV* separated by two vacuum pumps. The "Output flow valves" followed by mass flow controllers, control the output flow rate from the second set of buffers to the gas analyzers. Finally, samples can optionally be forwarded to a set of mass flow meters used for calibration purposes. The colored bottom bar below shows the range of pressure values at each stage.

The STEA sampling inlets were installed near the sonic's center of measurement volume. The horizontal separation was 155  22 cm, while the vertical separation between the two inlets was 2 cm. Upon sampling, the collected samples were carried using 6 mm Teflon tubes to the first set of buffers. The sampling can be summarized in the following steps (see a detailed description of the system operation and sampling in (Siebicke and Emad, 2019)):

1. 3D wind measurements are acquired from the sonic anemometer (uSonic-3 Class A) with a 10 Hz sampling frequency.

2. Wind coordinates are rotated into the streamline coordinates using the planar fit method without an intercept (Dijk et al., 160  2004). The fit is performed online as a running window operation with a window width of 2 days and an update frequency of once every 30 minutes.

3. The mean vertical wind from the previous 30 minutes interval is removed to minimize $\overline{w}$. This is equivalent to applying a high-pass filter to the vertical wind velocity measurements.

4. The active sampling line is determined (updraft or downdraft) based on the direction of the rotated vertical wind velocity 165  component.

5. The sampling scaling factor $A_i$ is calculated based on wind conditions in the near past and the calibration coefficients of the mass flow controllers. The scaling factor should be constant during the short accumulation intervals.

6. Air samples are collected, the controllers are adjusted to collect an air sample with a volume equal to $A_i\,|w|$.





7. When enough sample volume is accumulated in the respective buffer volume, samples are forwarded to the gas analyzer
for analysis. The amount of sample volume needed is determined based on the required flow rate for the gas analyzer
and the time needed to flush the tubes and the measurement cell and to perform enough repeated measurements.

8. Mean concentrations of accumulated samples are measured. The slow gas analyzer (LGR FGGA-24r-EP) alternates
on measuring the concentrations $C_i$ of the accumulated samples for updraft and downdraft. The accumulation time
for the short intervals was set to a fixed interval of one minute instead of an adaptive interval duration. During each
short interval, the gas analyzer performs repeated measurements for the gas concentration. The observed variability
for repeated measurements in the short averaging intervals was SD = 0.501 ppm which was similar to the measured
repeatability of the gas analyzer for a similar time interval.

## 2.5 STEA flux computations

This section describes the steps followed to obtain the final and corrected STEA flux. Firstly, we discuss the effect of water
vapor on the measured concentrations of other scalars and how we corrected that remaining water cross-sensitivity. Then, we
present the procedure of data quality screening. Next, we detail the steps of calculating the final STEA flux. Finally, we present
the buffer volume empirical correction we applied.

### 2.5.1 Water vapor correction

The gas analyzer used for the STEA measurements (LGR FGGA-24r-EP) reports the molar fraction of $CO_2$ and $CH_4$ of moist
air in parts per million (ppm). The measurements of $CO_2$ can not be used directly, as they are affected by the presence of water
vapor. The presence of varying water vapor concentrations in the sample affects the measurements of $CO_2$ and $CH_4$ in cavity
ring-down spectroscopy instruments in at least two ways: (i) the dilution effect, and (ii) the spectroscopic line broadening
(Rella, 2010). Rella (2010) proposed a quadratic equation to correct for the combined effect of line broadening and water
vapor dilution. The correction involves estimating the parameters ($a$) and ($b$) in the equation

$$r_c = \frac{\chi_c}{1 + a\,\chi_w + b\,\chi_w^2} \qquad (11)$$

where $r_c$ is the dry mole fraction of the species $c$, $\chi_c$ is the wet mole fraction measured by the instrument, and $\chi_w$ is the
water mole fraction measured by the instrument. For $CO_2$ measured by the LGR gas analyzer in ppm, Hiller et al. (2012)
experimentally estimated theses coefficients as, $a = -1.219 \times 10^{-06}, b = 1.229 \times 10^{-12}$.

We found that using the same parameters could not control for all the effects of water. A linear slope different from zero
was still found when supplying the gas analyzer with air of varying water concentration and of constant $CO_2$. This suggests
a remaining cross-sensitivity of $CO_2$ to the presence of water vapor. To control for this small remaining cross-sensitivity we
used a linear fit to obtain the slope and used it for the correction.

We were not able to supply the gas analyzer with air of known $CO_2$ concentration and varying water vapor. Instead, we used
the system's buffer volumes to collect and pressurize air from the atmosphere, closed the inlets, and supplied the gas analyzer





with enough sample flow rate for measurement. This procedure utilizes the effect of air drying due to decompression to deliver a varying water vapor content. Starting from humid atmospheric air near saturation ($RH \approx 90\%$, $T = 21$ °C). Air is sucked and compressed into the stainless steel buffer volumes to a pressure of $2.6$ bar. The water partial pressure in the pressurized buffers volumes will become higher than the saturation vapor pressure and water will precipitate leading to dryer air. Air is then decompressed and forwarded to the analyzer. As the buffer pressure is reducing, water vapor content will increase to reach

the same level of atmospheric humidity. Using this method we were able to modulate the water content in air from 6000 to 14000 ppm.

The accumulated sample was enough to supply the gas analyzer for ca. 10 min. We repeated the measurements several times and used the obtained dataset for correcting the remaining cross-sensitivity using a linear fit.

### 2.5.2 Raw data quality screening

Raw measurements of the wind velocity and scalar concentration were screened for outliers due to measurement errors and instrument malfunction. This included the following steps

- Statistical screening: despiking, dropouts removal (Vickers and Mahrt, 1997), and plausibility limits of raw gas analyzer and wind measurements (Sabbatini et al., 2018).

- Flushing time removal: measurement of the short interval events involve regularly switching the sampling line coming
to the gas analyzer between updraft and downdraft reservoirs. This caused subsequent samples to get contaminated. We experimentally chose a 25-second threshold at the start of each short interval event to account for the flushing time. The measurements falling before the threshold were discarded. Figure 4 shows an example of discarded flushing times at the start of each averaging interval.

- Detection of sample contamination: periods where the flow rate to the gas analyzer is smaller than $400$ mL min$^{-1}$ are
flagged. Under these conditions, ambient air might enter the system and contaminate the collected samples. When the number of flagged data points exceeds $10\%$ of the total points in the sampling interval, data in the sampling interval are discarded.

### 2.5.3 STEA flux calculation

After measurements are quality checked and erroneous data points are excluded, the final STEA flux is calculated as follows

- Short interval statistics: for each short interval sample, the gas analyzer will have several repeated measurements for the concentrations $C_i$, however, only one value is needed for the flux calculation. We use the median to obtain the representative value in order to minimize uncertainty and exclude outliers. Figure 4 shows an example of data quality checking and choice.

- Calculate air molar volume: the molar volume of air is needed to express the flux in units of $\mathrm{mol\,m^{-2}\,s^{-1}}$. The molar
volume is calculated using sonic temperature, pressure, and humidity measurements.





- – Calculate short intervals weights: following Eq. (6), the measured short interval concentration should be weighted by the ratio of the accumulated volume during that interval to the total buffer volume.

- – Calculate values of $\alpha_\theta$: values of $\alpha_\theta$ are calculated using vertical wind velocity and sonic temperature measurements. Values of $\alpha_\theta$ larger than 1 are discarded as they indicate a problem with the measurement.

- – Calculate updraft and downdraft mean concentrations: $C^\uparrow_{acc}$ and $C^\uparrow_{acc}$ are calculated for the averaging period $\Delta t$.

- – Calculate the flux: the STEA flux equation shown in Eq. (7) is used to obtain the final flux.

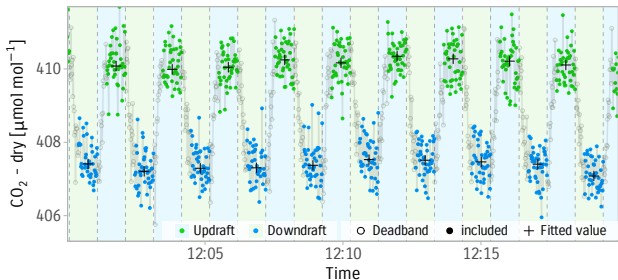

**Figure 4.** Data choice and fitting procedure for STEA method. Points represent consecutive concentration measurements from the gas analyzer. Updraft and downdraft samples are highlighted with blue and green, respectively. Grey hollow points are excluded from the data fitting (flushing time). Cross points are the chosen representative concentrations for each short interval (the median). Further quality checks for raw data are outlined in Section 2.5.2. Data are from 21 June 2020 at mid day.

### 2.5.4 Buffer volume empirical correction

Buffer volumes act on the signal as a low pass filter and introduce systematic bias to the fluxes. We used Eq. (10) to estimate the time constant of the buffer volumes used in our experiment. For each of the buffer volumes, a measurement point is acquired
every two minutes. The mean dimensionless mass flow rate to the gas analyzer was estimated from the pressure, the volume, and the estimated volumetric flow rate to the gas analyzer. We simulated the effect of buffer volumes on the high-frequency sonic temperature signal and parameterized the flux loss by artificially degrading the sonic temperature in a procedure similar to Goulden et al. (1996) and Berger et al. (2001)

### 2.6 EC reference flux measurements and computations

The raw data from the two sonics and the high-frequency gas density measurements from the IRGA were used to compute eddy covariance fluxes for water vapor and $CO_2$ in the period from 1 April 2020 to 1 November 2020 using EddyPro® software (LI-COR Env. Inc. USA) version 7.0.4. The flux processing steps were chosen to be as similar as possible to the TEA processing scheme. The calculation of EC fluxes involved: statistical screening for the data quality issues following (Vickers and Mahrt, 1997), mean removal by block averaging, compensation of the time lag between the wind and the scalar time series using





covariance maximization, tilt correction using the planar fit method without an intercept (Dijk et al., 2004) similar to TEA, and analytical high and low-frequency corrections to correct for the spectral attenuation of the IRGA (Moncrieff et al., 2005, 1997).

### 2.6.1 Density fluctuations correction

Due to using a closed-path gas analyzer, the TEA and STEA methods do not require WPL correction (Webb et al., 1980). WPL accounts for the effect of density fluctuations due to changes in temperature, humidity, and pressure. In TEA and STEA, after samples are collected and mixed in buffer volumes, the mean mixing ratio is measured. Therefore, no correction for density effects is needed. The measured TEA and STEA flux is equivalent to the flux measured with mixing ratios $\overline{r'_c w'}$.

### 2.7 Data selection for method comparison

For comparing the fluxes calculated from both methods, we selected averaging intervals according to the following criteria:

- Spike removal: following Vickers and Mahrt (1997) using a window width of 6 hours and a threshold of 2 standard deviations. This was mainly to account for unreliably elevated $CO_2$ concentration recorded by the open path gas analyzer due to water condensation.

- Rainy periods exclusion: data records during rainy weather conditions were excluded.

- Flux quality flags: periods where the flux quality flag is 1 or 2 according to Foken et al. (2005) were excluded.

- STEA low flow rate: averaging intervals flagged with the low flow rate flag described earlier were discarded.

After applying the above criteria, 992 averaging intervals remained. They accounted for $54.4\%$ of the whole comparison period. Nighttime data were the majority of excluded values. only 33% of averaging intervals were valid during night-time compared to 70% during daytime. The open-path gas analyzer used for EC produced unreliable measurements during high humidity conditions at night due to water condensation. Table 2 shows a summary of data quality checks results.

To compare the overall difference between the two methods, we used the coefficient of determination $R^2$ and the slope of the orthogonal distance regression (ODR) (also known as major-axis regression and model II regression). ODR considers the errors in $x$ and $y$ as opposed to OLS regression which assumes the error in $x$ is negligible (Wehr and Saleska, 2017).

## 3 Results and Discussion

We first discuss the newly proposed short-time eddy accumulation method. Then, we discuss some results and aspects of the STEA flux calculations. Afterward, we present the flux intercomparison between STEA and EC. Finally, we discuss the effect of using fixed buffer volumes on the fluxes and the proposed empirical correction.





**Table 2.** Summary of data quality checks for STEA and EC fluxes used in the EC/STEA flux intercomparison showing for each criterion, the number of averaging intervals that were excluded and the ratio of the excluded averaging intervals to the total. Details on the criteria and the thresholds used are provided in Sect. 3.3

| Criteria | Averaging intervals | Ratio (%) |
|----------|--------------------:|----------:|
| Spikes | 3 | 0.2 |
| EC missing value | 16 | 0.9 |
| Technical failure | 38 | 2.1 |
| Rain | 91 | 5.0 |
| STEA low flow rate | 107 | 5.9 |
| Flux quality flag 2 | 195 | 10.7 |
| Flux quality flag 1 | 382 | 20.9 |
| OK data | 992 | 54.4 |

### 3.1 Short-time Eddy Accumulation

Using the STEA method reduced the dynamic range requirement for eddy accumulation sampling. For a short averaging interval of one minute, the range was on average $60\%$ of the range required for the conventional eddy accumulation. As a result, the upper bound of the required dynamic range for $w$ reported by Hicks and McMillen (1984) as $5\sigma_w$ is lowered to

$3\sigma_w$. The reduction of the required dynamic range improves the accuracy and performance of the STEA system.

The accumulation on shorter time scales brings many advantages. First, it allows adapting to the local range of vertical wind velocity values which improves the resolution and dynamic range of the system. This can be achieved by exploiting the autocorrelation of the wind velocity signal to predict a scaling parameter, $A_i$ better adapted to the local velocity field for each interval. For a short interval, the range that the sampling apparatus needs to cover will be on average smaller than the range of

the whole averaging interval.

Additionally, the accumulation on varying intervals means the measurement frequency can be adjusted to match that of the gas analyzer or the precision requirements. This can be useful for reactive species and other trace gases, where relatively fast gas analyzers are available but not fast enough for EC.

Figure 5 demonstrates how the method works. In this example, the high-frequency samples are collected at 5 Hz frequency

for a 30-minute long averaging interval. The averaging interval is divided into 30 short intervals with a duration varying from 70 to 190 seconds. The flux in this example equals $-14.24\,\mu\mathrm{mol\,m^{-2}\,s^{-1}}$.

Finally, the STEA method facilitates using the STEA system in a continuous flow-through mode using rigid reservoirs. The operation in flow-through mode requires two sets of buffer volumes in a series as shown in Fig. 3. Two buffer volumes for each sampling line. The ideal operation of such a system can be achieved as follows:





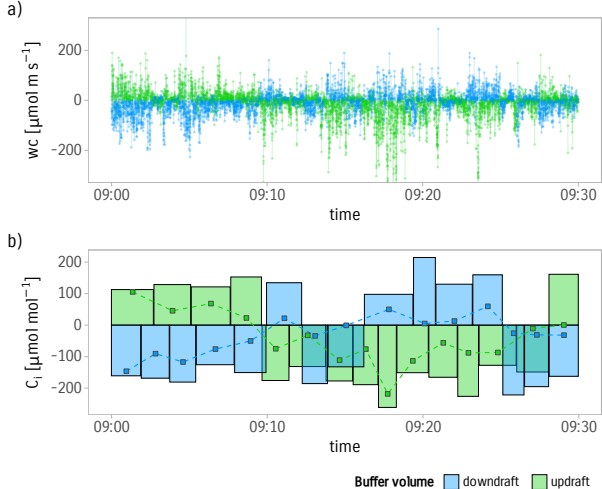

**Figure 5.** Sample accumulation using the STEA method. An example of 30 minutes of measurements: A) samples $wc$ are collected based on wind direction and proportional to its magnitude; B) Short intervals are accumulated. The variable short interval duration guarantees equal accumulated volume for consecutive short intervals. Points are the concentrations $C_i$ measured by the gas analyzer. The area of each rectangle represents the accumulated sample volume in arbitrary units and is equal to the relative weight for each concentration measurement. The sum of all measurements $C_i$ weighted by the relative sample volume will equal the covariance. Data are from 20 June 2020.

1. Wind velocity is measured, rotated, and the value of the scaling parameter $A_i$ is updated based on wind statistics and the flow calibration parameters.

   2. For each sampling line, air samples are collected into the respective set of buffer volumes continuously according to the sign of the vertical wind velocity and proportional to its magnitude and the value of $A_i$ until a predefined accumulated volume is reached.

3. When the predefined accumulated volume is reached, the second buffer volume in the sampling line is disconnected from the first. Sample accumulation time, $\tau_i$ and accumulated mass are recorded. Then, samples are forwarded to the gas analyzer.

   4. The slow gas analyzer alternates on measuring scalar concentration for each interval $C_i$ from the second set of buffer volumes for updraft and downdraft.

It is important for this scheme to keep the mass flow rate of air from the second set of buffer volumes to the gas analyzer constant for consecutive short intervals since the model used to represent the buffer volumes in Eq 8 assumes the flow rate to be constant with respect to time.





### 3.2 STEA fluxes computations

In this section, we discuss some aspects related to the calculation of the STEA fluxes. We first discuss the effects of water
vapor on $CO_2$ concentration measurements. Then, we discuss the effect of coordinates rotation on the fluxes.

#### 3.2.1 Water vapor correction

Treatment of the residual cross-sensitivity of $CO_2$ on water vapor content using a linear fit produced a small slope of $-1.17 \times 10^{-4}$ shown in Fig. (6). Thus, a difference in water concentration of $4000\,\mathrm{ppm}$ between updraft and downdraft reservoirs, typically observed in extreme conditions, will lead to a difference on the order of $0.5\,\mathrm{ppm}$ for $CO_2$.

Applying the water correction using the quadratic fit and the slope correction reduced the magnitude of STEA fluxes in comparison to the direct calculation of mixing ratios. However, it improved the fit between the STEA and the reference EC flux (Slope decreased from 1.18 to 1.04, and $R^2$ increased from 0.80 to 0.86.)

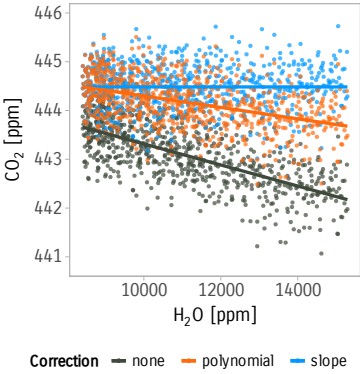

**Figure 6.** Effect of water correction on the measured $CO_2$ concentration using the LGR FGGA-24r-EP instrument. Points represent measured $CO_2$ by the gas analyzer when air with constant $CO_2$ concentration and varying $H_2O$ concentration was supplied. Lines represent linear regression fits. Red colored points and line represent $CO_2$ measurements after applying the polynomial correction (Hiller et al., 2012; Rella, 2010). In blue are the $CO_2$ measurements after applying our slope adjustment correction to remove additional cross-sensitivity on water.

#### 3.2.2 Coordinates rotation

The online coordinates rotation produced stationary rotation angles over the experiment period. The eddy covariance fluxes
calculated using the Class-A sonic using a two-month long dataset (1 June 2020 to 1 August 2020) produced the rotation angles: x-Pitch $= 0.6°$; y-Roll $= -4.3°$ (using the YXZ Euler convention). Whereas for the TEA moving-window online rotation, larger pitch angles were observed with a mean of $3.6°$ and values slowly climbing from 1.2 to 6° during the 6 weeks comparison period. The roll angle ranged from $-0.9°$ to $-0.24°$ with an average of $-0.4°$.





The use of online rotation with a moving window of two days minimized the residual mean vertical wind in comparison
to using the whole period of the experiment. This is likely due to a better adaptation to the local wind field. Furthermore, the
distribution of normalized mean vertical wind velocity of the short moving window had less spread, thinner tails and showed
more symmetry around the mean compared to the whole-dataset rotation. The residual mean magnitude of rotated $w$ for the
short moving window was $0.04\,\sigma_w$, the first and third quartiles were -0.03 and 0.03. Whereas for the whole-dataset rotation,
the mean magnitude was $0.17\,\sigma_w$ and the first and third quartiles were -0.07 and 0.22, respectively.

To estimate the effect of the online rotation method on the fluxes, we calculated EC fluxes using the two different rotation
approaches while keeping other treatments constant. The comparison revealed that the online rotation with a moving window
had minimal effect on the fluxes: a slope of approximately 1 and $R^2$ of 0.98 were obtained when using a linear fit. Nevertheless,
this comparison only included data of good quality from an ideal site. These results might differ for non-ideal conditions in a
more complex site.

## 3.3 STEA/EC flux intercomparison

The measured $CO_2$ fluxes using the STEA method in flow-through mode showed a good match with the reference EC fluxes
(Fig. (7)).

The time series of measured $CO_2$ fluxes in Fig. (7 - a) shows that the STEA method was able to reproduce the daily dynamics
of $CO_2$ flux very well. The estimated fluxes using the STEA method appear to have fewer spikes and smoother in general, this
is likely due to the smoothing effect of buffer volumes and the lower sensitivity of the closed-path gas analyzer to rain and high
humidity in particular during nighttime. The correction for nonzero mean vertical wind velocity using $\alpha_\theta$ was on average less
than 1.5% of the flux magnitude. This is due to the ideal topography of the site and the online rotation of the coordinate. The
correction at less ideal sites with more complex topography may differ.

The mean diurnal cycle estimates from the two methods match very well (Fig. (7 - b)). However, a small time shift can
be observed in the mean diurnal cycle as a result of the phase shift introduced by the low-pass filtering effect of the buffer
volumes.

The regression in Fig. (7 - c) shows that the measured $CO_2$ fluxes using the STEA method in flow-through mode have a
very good agreement with the reference EC fluxes. The magnitude of STEA fluxes was comparable to EC fluxes (ODR slope =
1.04). This indicates that the STEA method does not introduce systematic error to the fluxes. The coefficient of determination
$R^2$ was 0.86 which is not uncommon for careful side-by-side multi-method or even single-method flux measurements. The
remaining 13% of unexplained variance is the joint contribution by the uncertainties of the two flux estimates from the EC and
STEA methods. The observed uncertainty from the two methods calculated as the standard deviation of the difference was 4.36
$\mu mol\,m^{-2}\,s^{-1}$.

We suggest three different mechanisms contributing to the observed uncertainty leading to the unexplained variance between
the two estimates. First, the random sampling error arising from the stochasticity of turbulence (Hollinger and Richardson,
2005). The mean random sampling error of EC fluxes calculated following Finkelstein and Sims (2001) was $1.58\,\mu mol\,m^{-2}\,s^{-1}$.
The standard deviation of the difference between the two methods can be estimated to be $2.34\,\mu mol\,m^{-2}\,s^{-1}$ assuming the





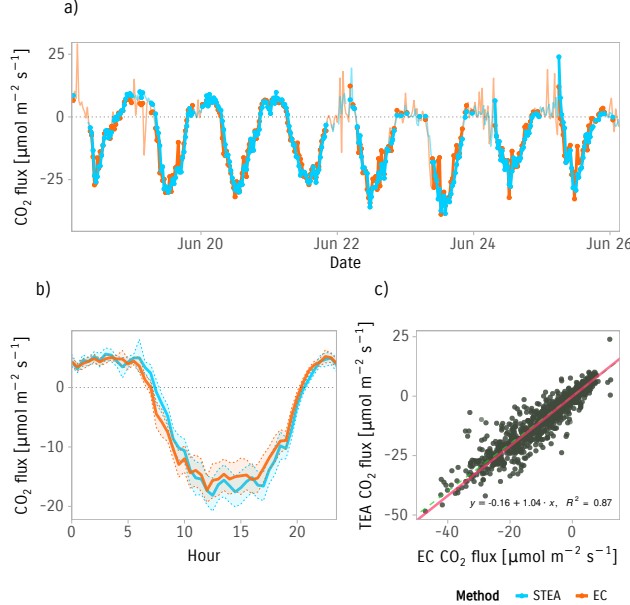

**Figure 7.** STEA and EC fluxes intercomparison. a) Time series of EC and STEA $CO_2$ fluxes for a subset period of 8 days. Points and thick lines indicate the averaging intervals used for comparison after filtering for quality. b) Mean diurnal cycle of $CO_2$ fluxes of STEA and EC. Bands are 95% confidence intervals of the mean calculated using nonparametric bootstrap. c) Scatter plot of STEA $CO_2$ fluxes against reference EC. The red line is the linear fit using the orthogonal distance regression (ODR). The dashed green line is a 1-to-1 line for reference.

STEA fluxes have a similar random sampling error. Therefore, the random sampling error of the two methods accounts for more than half of the observed variance. The difference between the two methods also shows heteroscedasticity with the er-
ror increasing along with the absolute magnitude of the flux, a similar behavior of the random sampling error was observed by Hollinger and Richardson (2005) when comparing two tower estimates. The second source of uncertainty is the use of different gas analyzers for STEA and EC. Polonik et al. (2019) compared five different analyzers for measuring $CO_2$ fluxes. They showed that the root-mean-square error (RMSE) was in the range of 1 to 3.35 $\mu mol\, m^{-2}\, s^{-1}$ depending on the analyzer type and the spectral correction method applied with larger discrepancies observed when comparing open-path to closed-path
sensors. Our results have an RMSE value of 4.39 $\mu mol\, m^{-2}\, s^{-1}$. While our result is slightly higher, it should be noted that RMSE is not an ideal metric for cross-studies comparison. A relative metric, such as $R^2$ would be more comparable but was unavailable. The third source of uncertainty is the use of buffer volumes in the STEA method. Figure (8 - a) demonstrates the increase of scatter in the measured fluxes due to the use of buffer volumes. Finally, the different processing steps between the two methods can contribute to the uncertainty. In particular, the effects of time-lag compensation, spectral corrections, and
statistical screening. We determined the combined effect of these processing treatments by calculating the EC flux with and without the treatments and found that the effect on the flux was negligible.





### 3.4 Effect of buffer volumes

Using fixed buffer volumes attenuates the signal. To understand the effect of buffer volumes use on the measured scalar

concentration, we carried out a simulation on a surrogate signal generated from sonic temperature. The simulation showed that

buffer volumes caused a decline that can reach up to 10% of the fluxes under operation ranges similar to those of our experiment

(for $\tau = 11$ minutes) (Fig. 8). The empirical correction was consistently able to mitigate most of the attenuation when the filter

properties are assumed to be constant, (i.e, the flow rate needs to be constant for consecutive short intervals). This assumption

was difficult to maintain using the 1-minute switching regime. The simulation showed the empirical correction for the buffer

volumes worked best when the correction factor was obtained using a linear fit, as opposed to taking a ratio of the attenuated

flux to the true flux for each averaging interval. The correction factor, in this case, is the reciprocal of the slope of the linear

regression between the attenuated flux and the true flux. The correction factor calculated using Eq. (8) shows a good agreement

between sensible heat flux and $CO_2$. However, the uncertainty of the correction factor increased with increasing buffer volume

time constant. For our experiment, the average time constant for the first-order linear filter used to model the buffer volume was

estimated to be $\tau = 700$ seconds. This value was used to simulate the loss on the fluxes using the sensible heat flux calculated

from the sonic anemometer. The correction factor was obtained from the slope of the attenuated flux and was equal to 1.18

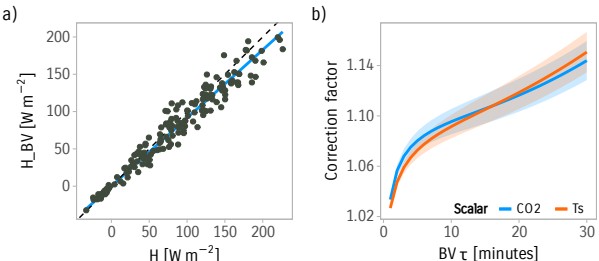

**Figure 8.** Empirical buffer volume correction. a) Effect of buffer volume attenuation on sensible heat flux with a time constant $\tau = 11$ minutes. The blue solid line is the linear fit between the two. b) Empirical correction factor for the effect of buffer volumes calculated as the reciprocal of the slope of attenuated flux for $CO_2$ and sensible heat flux. Bands are the estimated slope $\pm$ one standard error of the slope.

### 4 Conclusions

In this paper, we proposed an alternative method for the measurement of ecosystem-level fluxes. The new method, referred

to here as short-time eddy accumulation (STEA), allows the sample accumulation to be carried out on shorter varying-length

intervals. The STEA method offers more flexibility than the traditional TEA method and has many potential benefits. Most

importantly, STEA provides a higher dynamic range and better accuracy than the TEA method. It enables operating sample

accumulation under a flow-through scheme using fixed buffer volumes. The flexibility introduced by the STEA method offers



new ways to design eddy accumulation systems that are particularly suited for specific atmospheric constituents gas analyzers. For example, the accumulation time can be tailored to measure reactive species with lifetimes shorter than a conventional flux integration interval or to distribute the gas analyzer time to measure fluxes at different heights.

Furthermore, we presented a prototype evaluation of the STEA method under the flow-through regime. We described the details of the system design and operation. We compared flux measurements from our new system against a reference EC system over a flat agricultural field. The fluxes from the two methods were in very good agreement. We highlighted the importance of different processing and design aspects between the two methods and their potential effects on the fluxes.

Finally, we analyzed the effect of buffer volumes in the flow-through operational mode on the fluxes and proposed an 400 empirical correction to correct for the underestimation resulting from the low-pass filtering behavior of the buffer volumes.

In summary, the new STEA method provides a direct flux measurement method that complements the state-of-the-art EC method. It extends the coverage of micrometeorological methods to new trace gases and atmospheric constituents beyond the scope of the EC method.

*Code and data availability.* All data needed for producing the figures presented in the paper are provided at Emad and Siebicke (2021b). 405 Scripts for producing the plots in the paper are available at Emad and Siebicke (2021a). Currently, drafts are accesible at: https://s.gwdg.de/R4Fdhg and https://s.gwdg.de/CZ4zXI.

*Author contributions.* AE developed the theory of the STEA method and the empirical correction for the effect of buffer volumes, implemented needed software, performed the experiment, analyzed the data, interpreted the results, and wrote the manuscript. LS conceptualized the idea of flow-through eddy accumulation system, build the TEA system used in the experiment, planned and supervised the experiment, 410 provided feedback on the results, the analysis, and the manuscript.

*Competing interests.* The authors declare that they have no competing interests.

*Acknowledgements.* We gratefully acknowledge the support of the Bioclimatology group, led by Alexander Knohl, University of Göttingen, in particular technical assistance by Justus Presse, Frank Tiedemann, Marek Peksa, Dietmar Fellert, and Edgar Tunsch. We thank Christian Brümmer, Jean-Pierre Delorme from the Thünen Institute for Agricultural Climate Protection, and Mathias Herbst from the Center for 415 Agrometeorological Research of the German Meteorological Service (DWD) for facilitating the field work in Braunschweig. We further acknowledge Christian Markwitz for the fruitful discussions during the preparation of the manuscript and for reading and commenting on the manuscript. We thank Alexander Knohl, Nicolò Camarretta, Justus van Ramshorst, and Yannik Wardius for reading the manuscript and providing useful comments.



*Financial support.* The study was financially supported by the Ministry of Lower-Saxony for Science and Culture (MWK), by the European
Research Council under the European Union's Horizon 2020 research and innovation programme (grant agreement no. 682512 - OXYFLUX),
and by the Deutsche Forschungsgemeinschaft (INST 186/1118-1 FUGG).

## Appendix A:  List of symbols

**Table A1.** Symbols and subscripts with units

| **Symbols** | | |
| --- | --- | --- |
| $c$ | $\mathrm{mol\,m^{-3}}$ | Molar density of a scalar |
| $w$ | $\mathrm{m\,s^{-1}}$ | Vertical wind velocity |
| $\Delta t$ | s | Flux averaging interval |
| $A$ | – | TEA sampling scaling factor |
| $V$ | $\mathrm{m^3}$ | Volume |
| $C$ | $\mathrm{mol\,m^{-3}}$ | Mean concentration of accumulated samples |
| $\alpha_c$ | – | Transport asymmetry coefficient for scalar $c$ |
| $\rho$ | – | Correlation coefficient |
| $\dot{q}$ | – | Dimensionless mass flow rate |
| $\tau$ | s | Time constant of the buffer volume |
| $r_c$ | ppm | Mixing ratio in dry air for a scalar, $c$ |

| **Subscripts** | |
| --- | --- |
| $acc$ | Accumulated samples |
| $\uparrow$ | Updraft buffer volume |
| $\downarrow$ | Downdraft buffer volume |
| $c$ | Atmospheric constituent |



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
