# Peer review of "True eddy accumulation - Part 2: Theory and experiment of the short-time eddy accumulation method"

_Atmospheric Measurement Techniques, 2022_

## Author Comment (AC2)

**Authors' reply to reviewer comments 2**

Anas Emad and Lukas Siebicke

Dec 5, 2022

Review of the manuscript amt-2022-164 , "True eddy accumulation - Part 2: Theory and experiment of the short-time eddy accumulation method" by Emad and Siebicke (2022)

**General overview**

This, rather more practical and technology-oriented submission presents the second part of the two-part manuscript. The authors work through the mechanical arrangements of the TEA system and describe the processing and correction steps that were performed on the way from raw data to the final TEA fluxes. The field site and equipment is first described. The effects of buffer volume correction, coordinate rotation, data quality filters are discussed among other particularities. The performance of the TEA system is compared with that of a conventional EC setup. The work is detailed, interesting, and definitely deserves publication. I note that most of the previous comments have been taken into account by the authors, which leaves a relatively short list of corrections and additions provided below.

Recommendation: accept after the below minor corrections are done

**General comment**

We thank the reviewer for the positive review and the useful comments.

Below we have addressed the comments and suggestions.

- Our answers and comments are in blue.
- Summary of changes to the manuscript is in orange.

**Detailed comments**

Abstract, line 11: you can tell exactly how much the performance is improved (in relation to eddy-covariance).

> The performance improvements are expected in comparison to the conventional eddy accumulation. We will clarify this distinction.
>
> *Changes:*
> Added the phrase "compared to conventional TEA" to clarify where the perforamcnce improvments are.

Introduction/Theory sections: You briefly mention the accompanying paper on the lines 57-58, but I believe it should be given more space, perhaps in the Introduction, to explain the relationship between the two papers and how the current paper supplements Part I.

> The two papers are related in that they both address issues related to the absence of high-frequency information which is an intrinsic limitation in eddy accumulation. However, we believe that they form independently self-contained units. The decision to divide them into two papers was based on the topics they address, rather than a need for the second paper to build upon the first.

*Changes:*

We added a reference to the abstract to highlight the first paper and added more context to the few places where the first paper is mentioned.

line 75-76: "… limits the dynamic range and flexibility" is not immediately clear – can you specify what problems this entails, exactly. The paper is of technical character and I believe such detail should be added.

The limitations are mainly two points: i) longer averaging periods means the sampling appartus need to cover larger range of wind velocities and ii) the longer time is more prone to nonstationarities.

*Changes:*

Added a more detailed description of the limitations of the conventional TEA in section 2.1.

78: shorter intervals – corresponding to the individual updrafts/downdrafts?

Shorter interval are not necessarily corresponding to individual updrafts/downdrafts. They can be of any length.

*Changes:*

We have added a clarification to the nature of conditioning and improved the notation.

81-83: Not clear – basically this tells about two consequtive filters, both based on vertical wind velocity.

The method adds an additional conditional criterion (time) to the partition the random variable $wc$. The conventional TEA method is based on one conditional criterion (vertical wind velocity). Here, we additionally use short time intervals of variable length as a second conditional criterion.

*Changes:*

We added an additional sentence to clarify how the conditioning with time intervals works.

101-102: comment on the physical sense of the transport asymmetry coefficient.

Added a comment on the nature and estimation of the transport asymmetry coefficient.

110: please change the wording in "difference equation"

*Changes:*

Changed to "linear difference equation" to be more precise.

119: "…dimensionless flow rate" is unclear, and it continues to be unclear in Figure 1. Please briefly explain in the text how that is defined.

The mass in the buffer volume is conserved. Any air mass input to the buffer volume will have an equal mass flow exiting the buffer. The ratio of the mas of the input sample of air to the total mass of air in the buffer volume is the dimensionless flow-rate. It is essential to formulate the flow-rate in this way to construct the buffer volume model as a linear difference equation.

*Changes:*

We added an additional explanation to the definition of the dimensionless flow-rate in the text.

Figure 2: As I have noted earlier, it would be good to state in the caption that the photo does not reflect the state of vegetation during the experiment.

*Changes:*

We added a note indicating that the state of vegetation in the picture does not represent the state during the measurements.

Section 2.3.2: you should add some information on weather and other environmental conditions which prevailed during the period the measurements were active.

We added a description of the weather and environmental conditions in the text and a new figure of the main meteorological variables.

*Changes:*
We added a new figure of the main meteorological variables and a description of the weather and environmental conditions in the text.

Figure 3: explain MFC, MFM and VS in the caption

We added an explanation of the abbreviations in the caption. VS was a typo, we removed it.

201-202: please revise the sentences.

Sentence were revised and changed.

226-227: is the median really better than mean? Are the outliers that important for the mean? The outliers can in some cases be related to strong source/sink hotspots.

The choice of the median is not expected to have an effect on the final flux as the repetitions of $C_i$ represent measurements of the same quantity and are not expected to vary. The median was slightly better than the mean in few cases where flow-rate instationarities created large spikes.

233: remind here what alpha was.

*Changes:*
We added more information about alpha and referenced the accompanying paper.

Coordinate rotation in a flat site

317: are the parameters correct?

The parameters are correct. Water vapor cross-sensitivity is a large source of bias in CRDS systems. This was particularly problematic for TEA as for most time of the day there is a large difference in the concentration of water vapor between the updraft and downdraft reservoirs.

319: "stationary rotation angles" is not clear

*Changes:*
We changed the wording to "rotation angles with low variability".

Please do not start sentences with "whereas"

*Changes:*
The sentence was rephrased.

Section 3.2.2: so I understand that the difference between the two runs of coordinate rotation results from the different methods used, as the same anemometer (?) was shared the EC and TEA systems?

Furthermore, I wonder how the angles of 4-6 degree angles could be observed in a seemingly flat site – please comment on that.

Indeed the same anemometer was used for both systems. However, the differences are due to applying the planar fit routine as a running window with a rather short window of two days in TEA. This suggests the differences are due to temporal wind circulation patterns and not due to a physical angle.

We believe 4-6 degree angles are not uncommon in flat sites given the possibility of a tilted instrument.

350: clarify "uncommon"

*Changes:*
The sentence was removed as it was found to be unnecessary.

344-346: given the good match otherwise, I would suggest that this phase shift is the next target for correction.

The issue with correcting the phase shift was indeed a topic of a future study and was highlighted in the text.

Section 3.3: as an ultimate measure of similarity between EC and TEA flux, I would recommend presenting the cumulative Re and GPP for the study period, and present it in g C m-2, with uncertainty included. This would convince those users who are concerned with getting the correct integrated flux values (they are the majority).

We thank the reviewer for the useful suggestion. While we understand the value of adding plots for Re and GPP, we have decided not to include them in our current research for the sake of simplicity. We have found that the cumulative total flux provides a similar level of insight and allows us to focus on the most important aspects of our analysis which is evaluating the performance of the STEA method. We think that additional partitioning to Re and GPP, although very useful to understand the ecosystem, is unlikely to help in comparing the two methods.

*Changes:*
We added a new figure showing the cumulative flux for the study period and added our interpretation of the results.

Section 3.4. In the lines 344-346, you mention that the buffer volumes also lead to the phase shift, which results in an noticeable deviation the from the EC flux diurnal course. How does this relate to what is described in 3.4? Can these issues be addressed simultaneously, at least in theory?

We thank the reviewer for raising this important point. Indeed, a direct correction approach that accounts for the phase shift is possible and is presented in a future paper (currently available as a preprint).

*Changes:*
We added a new paragraph to refer to a direct correction scheme that we developed recently and is the topic of a future paper.

Conclusions: I think here, and also in the results section, you should estimate the effective time resolution of the TEA flux achievable with the presented technique. The community is more familiar with the EC technique and usually thinks in terms of 30 min average fluxes – how does TEA perform in that respect?

While it is indeed possible to calculate fluxes for shorter intervals with the new method, we chose the conventional 30-minute throughout the paper to be consistent with the EC literature. All short intervals are integrated to the 30-minute averaging interval and are not expected to be any different from calculating the flux directly from the raw data.

We think that there might be uses for calculating the fluxes on shorter time intervals apart from the practical advantages in STEA implementation which motivated our formulation. However, we think such uses would form a separate topic.